# Genetic Background of Epilepsy and Antiepileptic Treatments

**DOI:** 10.3390/ijms242216280

**Published:** 2023-11-14

**Authors:** Kinga Borowicz-Reutt, Julia Czernia, Marlena Krawczyk

**Affiliations:** Independent Unit of Experimental Neuropathophysiology, Department of Toxicology, Medical University of Lublin, Jaczewskiego 8b, 20-090 Lublin, Poland; julia.czernia@umlub.pl (J.C.); marlena.krawczyk1@umlub.pl (M.K.)

**Keywords:** gene variants, genotype–phenotype correlation, genotype–treatment response relationship, rare epilepsies

## Abstract

Advanced identification of the gene mutations causing epilepsy syndromes is expected to translate into faster diagnosis and more effective treatment of these conditions. Over the last 5 years, approximately 40 clinical trials on the treatment of genetic epilepsies have been conducted. As a result, some medications that are not regular antiseizure drugs (e.g., soticlestat, fenfluramine, or ganaxolone) have been introduced to the treatment of drug-resistant seizures in Dravet, Lennox-Gastaut, maternally inherited chromosome 15q11.2-q13.1 duplication (Dup 15q) syndromes, and protocadherin 19 (PCDH 19)-clusterig epilepsy. And although the effects of soticlestat, fenfluramine, and ganaxolone are described as promising, they do not significantly affect the course of the mentioned epilepsy syndromes. Importantly, each of these syndromes is related to mutations in several genes. On the other hand, several mutations can occur within one gene, and different gene variants may be manifested in different disease phenotypes. This complex pattern of inheritance contributes to rather poor genotype–phenotype correlations. Hence, the detection of a specific mutation is not synonymous with a precise diagnosis of a specific syndrome. Bearing in mind that seizures develop as a consequence of the predominance of excitatory over inhibitory processes, it seems reasonable that mutations in genes encoding sodium and potassium channels, as well as glutamatergic and gamma-aminobutyric (GABA) receptors, play a role in the pathogenesis of epilepsy. In some cases, different pathogenic variants of the same gene can result in opposite functional effects, determining the effectiveness of therapy with certain medications. For instance, seizures related to gain-of-function (GoF) mutations in genes encoding sodium channels can be successfully treated with sodium channel blockers. On the contrary, the same drugs may aggravate seizures related to loss-of-function (LoF) variants of the same genes. Hence, knowledge of gene mutation–treatment response relationships facilitates more favorable selection of drugs for anticonvulsant therapy.

## 1. Introduction

Epilepsy is a leading neurological disorder, affecting 40–60 million people worldwide. Only correctly selected treatments can provide patients with a seizure-free life. Unfortunately, despite the use of rational polytherapy based on epilepsy symptomatology, refractory seizures are still diagnosed in approximately 25–30% of patients. Traditional anticonvulsant treatment is based on medications affecting the function of ion channels and receptors that are substantially involved in the balance between excitation and inhibition processes in the brain. Among the available antiepileptic drugs, we can find variety of sodium and calcium channel blockers, potassium channel openers, GABA_A_ receptor agonists, and glutamate receptor antagonists [1,2].

Genetics is increasingly entering the field of epileptology. Genetic factors play a significant role in the pathogenesis of epilepsy, with studies indicating that approximately 50% of cases have a genetic background. In the last decade, numerous mutations causing epilepsy have been described, improving our understanding of the molecular mechanisms underlying clinical manifestation of congenital epilepsies. This gave hope that the determination of gene variants, also in patients with acquired polygenic epilepsy, may open the way to targeted personalized therapy and a reduction in drug resistance. Another possible advantage may be greater homogeneity of patients (carrying the same defective gene variants) recruited to clinical trials, which should provide greater reliability of the obtained results. In the future, genetic epilepsy, particularly monogenic epilepsy, could be subjected to precision therapy, including genotherapy [1,3].

The most common mutations occur in genes controlling processes of neuronal metabolism and excitability, cell signaling pathways, and synapse formation. Importantly, mutations within the same gene may induce the synthesis of different pathogenic protein variants, thus resulting in heterogeneous clinical epilepsy phenotypes and often different drug responses. For instance, sodium channel blockers are effective in patients with (GoF mutations in the *SCNA2* and *SCN8A* genes. In contrast, this class of antiepileptic drugs can aggravate seizures caused by LoF variants in *SCNA1* and *SCN2A*. All the mentioned genes encode for sodium channel subunits. On the other hand, seizures caused by GoF mutations in the *GRIN2A* gene, encoding for the GluN2A subunits of NMDA receptors, can be treated with memantine, an NMDA receptor antagonist that was previously used in the treatment of Alzheimer’s disease. This is an example of so-called drug repurposing, where old existing drugs are used in new therapeutic indications. The same is true for quinidine, an antiarrhythmic drug and a partial antagonist of KNa1.1 channels that reduce seizures related to GoF mutations in *KCNT* genes, which encode potassium channel subunits [1,4,5].

In this article, we focus on the most common mutations in genes encoding sodium and potassium channels as well as GABA_A_ and glutamate ionotropic receptors. Special attention will be paid to the genotype–phenotype correlations and genotype–drug response relationships.

## 2. Mutations in Genes Encoding Sodium Channel Subunits

Voltage-gated sodium channels are involved in the generation and development of neural action potentials. Mutations in the *SCN1A*, *SCN1B*, *SCN2A*, *SCN3A*, and *SCN8A* genes are responsible for the development of refractory genetic epilepsies in the beginning in infancy or childhood. In this subsection, we focus on the three genes (*SCN1A*, *SCN2A*, and *SCN8A*) which have the greatest clinical implications.

### 2.1. SCN1A Gene Mutations

The sodium channel Na_v_1.1 subunit, encoded by the *SCN1A* gene, is predominantly expressed within inhibitory neurons. *SCNA1* is located on chromosome 2q24. Approximately 1200 different mutations in this gene, including nonsense, missense, and frameshift variations, have been identified so far. Larger modifications such as deletions are also possible, particularly when the entire gene or its specific regions are involved [5,6].

*SCN1A* mutations can result in a wide range of seizures, including Dravet syndrome and other pediatric epilepsies of variable severity, like generalized epilepsy with febrile seizures plus (GEFS+). The remaining phenotypes related with these mutations include familial simple febrile seizures, focal epilepsies, or early infantile epileptic encephalopathy. It should be noted, however, that the identification of a specific variant does not reliably predict how serious the development of clinical consequences will be. *SCN1A* gene mutations passed on by parents cause less severe seizures than de novo variants that emerge for the first time in a given patient. It is worth remembering is that de novo mutations constitute approximately 80% of all cases [5].

Dravet syndrome, also termed severe myoclonic epilepsy of infancy (SMEI), was first documented by Charlotte Dravet in 1978. It is an example of a drug-resistant epileptic encephalopathy. This syndrome is most often the result of a LoF mutation in the *SCN1A* gene. Disruptions in the *SCN9A*, *SCN2B*, *PCDH19*, *GABRG2*, *GABRA1*, and *STXBP1* genes can contribute to Dravet syndrome less frequently (the occurrence rate was estimated to be approximately 15.9–18% of the population). In this condition, children, usually at the age of 4–8 months, experience mostly tonic–clonic and myoclonic seizures, which are commonly triggered by fever. However, Dravet syndrome is characterized by a very heterogeneous symptomatology. The first symptoms include seizures, drooling, loss of consciousness, rapid eye movements, temporary apnea, myoclonic jerks, and possibly early photophobia. The clinical manifestation of seizures is very diverse and includes febrile and afebrile, generalized and unilateral, and clonic and tonic–clonic convulsions. Other forms of seizures, such as myoclonic, focal, and unconscious seizures, start to emerge between 1 and 4 years of age. Walking difficulties, behavioral problems, and cognitive decline are common accompanying disorders. Patients with less distinct clinical features who lack myoclonic seizures, generalized spike and wave activity, loss of consciousness, and photophobia are diagnosed with borderline Dravet syndrome or SMEI-borderline (SMEB) [7,8,9,10].

Treatment of Dravet syndrome presents a real challenge, while status epilepticus, frequently occurring in this condition, is considered to be exceptionally drug resistant. Experimental studies have revealed that LoF *SCN1A* mutations impair the function of inhibitory, but not excitatory, neurons in several seizure-related brain regions. Therefore, sodium channel blockers, including phenytoin, carbamazepine, oxcarbazepine, or lamotrigine, are not only ineffective, but they even aggravate convulsions. Among the more effective drugs are valproate, benzodiazepines, stiripentol, and cannabidiol. Also, a ketogenic diet can alleviate seizures in Dravet syndrome. Currently, a combination of stiripentol, valproate, and clobazam, or alternatively, valproate with topiramate, are the approved treatment procedures. Interestingly, combined treatment with valproate and stiripentol is more effective in Dravet syndrome related to *SCN1A* mutations compared to mutations in other genes. Moreover, therapy with stiripentol is more successful in patients with missense *SCN1A* mutations than in those with truncated *SCN1A* mutations [1,11].

Cannabidiol, approved as an antiseizure drug in Dravert syndrome, has been linked to a higher occurrence of side effects. Vomiting, exhaustion, fever, upper respiratory tract infection, decreased appetite, convulsions, lethargy, somnolence, and diarrhea are listed among the frequent adverse effects that occur in more than 10% of cases. Another hopeful drug seems to be soticlestat, a novel cholesterol 24-hydroxylase inhibitor. In a recent clinical study of ELEKTRA, adjunctive treatment with soticlestat showed a significant 13% reduction in seizure frequency with a good safety profile. On the other hand, fenfluramine, an amphetamine derivative used previously as an anti-obesity drug, was withdrawn from the market due to serious cardiovascular side effects. This drug, applied at lower doses, shows encouraging effects in the treatment of Dravet syndrome-related seizures. Importantly, this low-dose treatment does not result in the development of valvular heart disease or pulmonary arterial hypertension. 

Another interesting substance is ganaxolone, a brand-new medication for people with cyclin-dependent kinase-like 5 (CDKL5) deficiency disorder. This neurosteroid, being a positive allosteric modulator of GABA_A_ receptors, is also used in the treatment of epilepsy related to PCDH19 clusters, status epilepticus accompanying Dravet syndrome, and infant convulsions following hypoxic brain damage. The fact that this drug produces few to no negative effects on the developing brain makes it a near-ideal antiepileptic medication in children. Among the other medications approved for the treatment of Dravet syndrome are lorcaserin, LP352, BL-001, BMB-101 (all four are selective 5HT_2C_ receptor antagonists), NT102 (a vitamin K analog), and clemizole (an inhibitor of the transient receptor potential channel 5). Ongoing clinical trials are examining the effectiveness of benzodiazepines, alprazolam, and intranasal diazepam [12,13,14,15,16,17,18].

However, as with many other drugs in development, promising preliminary preclinical and early clinical data may be not predictive of real clinical benefits. One of the reasons for this may be the small size of the studied groups [3]. Either way, the results of the few abovementioned clinical studies should be interpreted with caution.

Modern strategies considered for the treatment of *SCN1A*-related seizures include the use of antisense oligonucleotides restoring the normal function of *SCN1A*; Hm1 peptide, a selective activator of Na_v_1.1 inhibitory interneurons; and gene therapy [1,3].

### 2.2. SCN2A Gene Mutations

*SCN2A* is a gene found on chromosome 2, and it is responsible for coding the alpha subunit of voltage-gated sodium channels (Na_v_1.2). These channels are predominantly present in neocortical and hippocampal excitatory neurons, and they induce rapidly activating and inactivating currents that support the generation of action potentials. *SCN2A* mutations may cause neonatal-, infantile-, and childhood-onset epilepsies. Initially, familial GoF mutations were found in patients with mild self-limiting and pharmacoresponsive seizures, known as benign familial neonatal–infantile epilepsy (BFNIE). Children with BFNIE generally do not show impaired intellectual development. Seizures are successfully controlled with carbamazepine, valproate, and phenobarbital. In contrast, recently discovered de novo *SCN2A* mutations were found to be the most frequent causes of developmental and epileptic encephalopathy (DEE), manifesting as autism spectrum disorder, intellectual disability, and severe refractory seizures. Their symptomatology may also include movement disorders, e.g., dystonia and chorea. Furthermore, severe clinical manifestations in the form of Ohtahara syndrome, West syndrome, Lennox–Gastaut syndrome, and infantile epilepsy with migrating focal seizures (EIMFS), have been linked to spontaneous (de novo) mutations in the *SCN2A* gene. These mutations are implicated in a specific form of early infantile epileptic encephalopathy, designated as type 11 [19,20].

Patients with *SCN2A*-related DEE and early infantile epilepsy respond well to sodium channel blockers, particularly phenytoin and carbamazepine. Sometimes, higher doses of phenytoin are required. The ketogenic diet and high-dose steroid treatment are also effective. Interestingly, patients with early-onset seizures present with missense GoF mutations. On the contrary, late-onset forms of epilepsy are associated with LoF truncation mutations in the *SCN2A* gene. In this case, the seizures worsened after sodium channel blockade. This proves that functionally different mutations in the same gene may lead to different therapeutic implications [21,22].

### 2.3. SCN8A Gene Mutations

The *SCN8A* gene, located on chromosome 12q13, encodes for Na_v_1.6 channel subunits. These channels are mostly expressed in excitatory neurons, and less frequently in inhibitory nerve cells. In general, GoF mutations seem to be responsible for *SCN8A*-related encephalopathy, with seizures beginning within the first 18 years of life. De novo missense GoF *SCN8A* mutations, which result in single amino acid alterations in the protein product, are linked to the severe form of early infantile epileptic encephalopathy 13 (EIEE13). Some electroclinical parameters of this disorder indicate a widespread gradual deterioration of brain function, mostly in the temporal and occipital lobes. Patients with *SCN8A* mutations develop seizures with different morphologies, including focal or afebrile generalized tonic–clonic convulsions, infantile spasms, and absence seizures, usually accompanied by psychomotor disorders of various degrees. The most common phenotypes include Lennox–Gastaut syndrome, West syndrome, and variety of epileptic encephalopaties. Again, GoF mutation-related seizures are controlled by treatment with sodium channel blockers, like phenytoin, carbamazepine, oxcarbazepine, lacosamide, lamotrigine, rufinamide, or topiramate. In contrast to GoF *SCN8A* mutations causing epilepsy, there are also LoF variants leading to cognitive disorders or autism without epilepsy [6,23,24,25,26].

Detailed genotype–phenotype correlations in patients with *SCN8A* mutations have been presented in six different clinical groups: BFNIE with normal cognition and pharmacoresponsive seizures, intermediate epilepsy with mild intellectual disability and partial pharmacoresponsiveness, DEE with severe intellectual disability and mostly pharmacoresistant seizures, generalized epilepsy with mild to moderate intellectual disability and absence seizures, unclassifiable epilepsy, and mild to moderate neurodevelopmental disorders without epilepsy. Two of the described mutations related to DEE showed strong GoF characteristics, while one variant causing BFNIE or intermediate epilepsy presented mild GoF effects. In contrast, three variants causing generalized epilepsy were of a LoF type. As might be expected, the GoF variants responded better to sodium channel blockers. However, there was a case of a female infant with DEE, in whom therapy with phenytoin, levetiracetam, valproate, or phenobarbital was entirely ineffective. In contrast, oxcarbazepine significantly reduced seizure frequency and abolished tonic convulsions. In another case of a 5-year-old patient with a GoF *SCN8A* mutation and encephalopathy, phenytoin maintenance therapy led to rapid deterioration, with encephalopathy, ataxia, and somnolence. Better control of the seizures was achieved when lacosamide maintenance therapy was introduced, and phenytoin was applied only as an emergency drug to stop exacerbated convulsions and prevent the development of status epilepticus. It seems that sodium channel blockers, including phenytoin, can be effective in patients with GoF *SCN8A* mutations; however, due to potential cognitive impairment that is undesirable in patients with encephalopathy, phenytoin should be considered as a last-resort drug [27,28,29,30,31].

Among the new therapeutic strategies, NBI-921352 (formerly known as XEN901), a selective Na_v_1.6 inhibitor, has been designed to treat early infantile epileptic encephalopathy resulting from GoF *SCN8A* mutations. Another possibility is GS458967, a potent sodium channel modulator [1].

Interesting effects were observed in a kainate model of mesial temporal lobe epilepsy. *SCN8A* gene expression in the mouse hippocampi was reduced by using a small hairpin interfering RNA directed against *SCN8A*. As a result, the development of spontaneous seizures and reactive gliosis were significantly reduced. This suggests that the selective targeting *SCN8A* gene may be useful in the treatment of temporal epilepsy [31,32]. Table 1 presents data on epilepsy related to mutations in genes encoding sodium channels.

## 3. Mutations in Genes Encoding Potassium Voltage-Gated Channels

Potassium channels are membrane-spanning proteins that are found in a variety of cells. In neurons, these channels modulate neuronal excitability and are involved in several processes, e.g., generation of action potentials, rapid conduction of high-frequency action potentials, quick axon repolarization, maintenance of resting membrane potential, regulation of neurotransmitter release, development of neuronal circuitry, and moderation of cell death/survival signaling pathways. It should be noted that the activation of potassium channels decreases neural excitability, leading to inhibitory effects. Thus, it is not surprising that most epileptogenic mutations in genes encoding potassium channel subunits have LoF characteristics. In fact, four functional subunits of potassium channels are encoded by more than 80 genes, with mutations leading to different neurological disorders: mainly epilepsy and developmental delays [33,34,35]. In the following subsection, we focus on the *KCNQ2*, *KCNQ3*, *KCNA1*, *KCNT1*, and *KCNT2* genes.

### 3.1. Mutations in KCNQ2 and KCNQ3 Genes

The *KCNQ* gene subfamily consists of five members (*KCNQ1*–*KCNQ5*), each encoding respective subunits of Kv7.2–5 voltage-gated potassium channels. Mutations in the *KCNQ* subfamily may lead to the development of congenital seizures such as benign familial neonatal seizures (BFNS) and early-onset epileptic encephalopathy (EOEE) [36,37,38,39,40].

*KCNQ2* is located on chromosome 20q13.3, while *KCNQ3* on chromosome 8q24. *KCNQ2* encodes four α-subunits of Kv7.2 potassium channels, which interact with the Kv7.3 α-subunits encoded by *KCNQ3*. Together, KV7.2/KV7.3 form the homo- and heterotetrameric voltage-gated potassium channels responsible for the M-current, a non-inactivating current that raises the threshold for firing an action potential. Both *KCNQ2* and *KCNQ3* are co-expressed in the same brain areas, which contributes to the development of similar clinical phenotypes related to specific mutations [40,41,42,43,44,45,46].

In more than 80% of patients, *KCNQ2* and *KCNQ3* variants are associated with benign familial neonatal epilepsy (BFNE). This relatively mild condition is typically manifested by clusters of focal seizures with alternating laterality, which begin in the first few days of life. *KCNQ2* and *KCNQ3* variants can also lead to severe refractory epilepsy accompanied by encephalopathy and cognitive decline. It was reported that up to 83% of the variants causing refractory epilepsy were found to be de novo. This remains in contrast to BNFE, where only a mere 8% of the variants were newly occurring mutations [1,41,42,46,47,48].

Epilepsies associated with *KCNQ2* mutations are recommended to be treated with sodium channel blockers and retigabine, a selective Kv7 channel activator. However, retigabine has been withdrawn from the market for safety reasons. Carbamazepine and lamotrigine are among the most effective antiepileptic drugs employed in the treatment of seizures related to *KCNQ2* LoF variants. The reason of so good response to sodium channel blockers remains unclear. It may be explained by co-localization of potassium and sodium channels and the presence of a common structural fragment in Kv7 and sodium channels. In cases when carbamazepine is ineffective, oxcarbazepine or barbiturates (e.g., intravenous thiopental) may be considered as alternatives. Another anticonvulsant conventionally used in patients with BFNE is phenobarbital, an agonist of the barbiturate site within the GABA_A_ receptor complex. On the other hand, some reports indicate that this drug is ineffective in the treatment of genetic epilepsies in newborns. Despite contradictory data, the World Health Organization still recommends phenobarbital as the first-line therapy for infantile seizures, irrespective of their underlying causes. Nevertheless, the potential negative long-term influence of barbiturates on the developing brain has prompted researchers to explore new therapeutic options. Presently, an investigational drug, XEN496 (a novel immediate-release formulation of retigabine), is in a Phase III clinical trial [1,48,49,50,51].

Although the most frequent *KCNQ2* and *KCNQ3* mutations are LoF, some *KCNQ2* variants causing encephalopathy may present GoF characteristics, with increased channel activity, altered kinetics, and more severe clinical manifestation in affected patients. Not surprisingly, in such cases, sodium channel blockers are not as effective as in LoF variants, while retigabine, as a KV7 activator, may even aggravate all clinical symptoms [1].

Recently, a missense LoF *KCNQ3* variant was discovered in patients with BFNE. This mutation results in decreased voltage sensitivity of channels. Interestingly, treatment with β-hydroxybutyrate (ketogenic diet) restored the normal function of the defective Kv7.3 S4 channel segment [1,45,47].

### 3.2. Mutations in KCNA1 and KCNA2 Genes

The *KCNA1* gene, located on chromosome 12p13.32, carries instructions for creating the α-subunit of Kv1.1 voltage-gated potassium channels. These channels exhibit the capacity to counterbalance the impacts of depolarizing input signals, thus preventing neuron overexcitation. Mutations in *KCNA1* are mostly associated with episodic ataxia type 1 (EA1). However, the development of epilepsy and paroxysmal kinesiogenic dyskinesia is also possible. EA1 is a result of hyperexcitability of cerebellar interneurons and manifests in the form of motor deficits. The main cause of EA1 is a missense LoF mutation in the *KCNA1* gene, which disrupts Kv1.1 function and excessively inhibits Purkinje cells. Patients with EA1 experience attacks of ataxia, myokymia, and dysarthria triggered by stressors. *KCNA1* mutations are reflected in the whole protein. Therefore, patients develop different clinical phenotypes and various concomitant disorders (e.g., hyperthermia or seizures). Genetic modifiers or environmental factors may also affect the symptomatology of the disease [52,53].

Antiepileptic medications, such as carbamazepine, phenytoin and lamotrigine, acetalozamide, and sometimes benzodiazepines, show positive effects in alleviating the symptoms in EA1. Another therapeutic option is niflumic acid, which enhances the activity of Kv1.1 and may ameliorate channel dysfunctions. However, due to the limited number of studies and trials that directly compare the effectiveness of these treatments, there is currently no conclusive evidence supporting the efficacy of any specific drug. The management of EA1 is still symptomatic and based on medications that restore normal neuronal excitability [54,55,56,57,58].

Interestingly, mutations in *KCNA1* are considered to be potential biomarkers for sudden unexpected death in epilepsy patients (SUDEP), thus further highlighting the prospective benefit of Kv1.1-targeted therapeutic approaches [59,60].

The *KCNA2* gene is located on chromosome 1.13.3. Kv1.2 channels, encoded by *KCNA2*, form homomers and heteromers in conjunction with Kv1.1 or Kv1.4. These channels conduct a voltage-dependent potassium “delay” (D-type) current. The D-type current is activated even below threshold; therefore, it delays the initiation of action potentials and prevents repetitive firing. *KCNA2* mutations manifest as eight rare genetic neurological movement disorders, e.g., EAs with dysarthria, nystagmus, or difficulties in performing purposeful movements. Mutations in the *KCNA2* gene are related to severe forms of epileptic encephalopathy. The course of the disease varies depending on the type of mutation. LoF variants result in focal seizures with prominent sleep activation. GoF mutations lead to more severe generalized seizures with ataxia and cerebellar atrophy. The most severe early-onset phenotypes, with generalized or focal seizures and developmental impairment, may be manifestations of either GoF or LoF mutations [57,59,60,61,62,63,64,65,66].

### 3.3. Mutations in KCNT1 and KCNT2 Genes

The *KCNT1* and *KCNT2* genes belong to the Slo2 family of genes encoding for Na^+^-dependent (K^+^) channel pore-forming α-subunits that can be regulated by changes in voltage, intracellular ions, and second messengers. The human *KCNT1* gene is located on chromosome 9q34.3 and encodes for a ligand-gated potassium channel known as “sequence like a calcium activated K^+^ channel” (Slack). This channel is activated by the intracellular sodium concentration. Expression profiling studies have revealed significant expression of this gene in most brain regions, with the exception of the corpus callosum and substantia nigra. All pathogenic variants discovered in the *KCNT1* gene have GoF phenotypes. Mutations in this gene are considered to be the most common genetic causes of autosomal-dominant nocturnal frontal lobe epilepsy (ADNFLE); encephalopathy with malignant migrating focal seizures of infancy (MMFSI), known as a form of EOEE, and the most severe, EIMFS in neonates and infants. Patients with ADNFLE experience violent motor sleep-related seizures. Epilepsy in these patients, especially those with EIMFS or MMFSI phenotypes, is often highly drug resistant and leads to severe developmental and functional disabilities [67,68,69,70].

Importantly, ADNFLE and nocturnal frontal lobe epilepsy (NFLE) phenotypes related to *KCNT1* mutations tend to be more severe, with refractory seizures and a higher prevalence of intellectual disability or psychiatric features when compared to ADNFLE caused by mutations in the *CHRNA4* and *CHRNB2* genes encoding for neuronal nicotinic receptor subunits (alpha4 and beta2, respectively). ADNFLE presents a different phenotype than MMFSI; therefore, it is supposed that both can be part of a larger spectrum of epilepsies related to this gene. Indeed, *KCNT1* mutations have been detected in individuals experiencing various types of EOEE, focal epilepsy, and leukoencephalopathy. Other variants of *KCNT1* have been linked to other epilepsy-related conditions, including West and Ohtahara syndromes, which are rare forms of severe epilepsy in neonates [71,72,73,74].

Conventional antiseizure medications have limited efficacy in the treatment of *KCNT1*-related epilepsies, which further reduces patients’ quality of life. A combination of stiripenol with benzodiazepines (commonly clonazepam, clobazam, or nitrazepam), levetiracetam, and the ketogenic diet have been well-tolerated, but with limited therapeutic effects. From a theoretical point of view, seizures caused by *KCNT1* mutations could be treatable with a drug specifically targeting *KCNT1* channels. Quinidine, an antagonist of various potassium channels, including KNa1.1, has recently gained attention as a potential treatment option. However, response to this drug is unpredictable, but much better when treatment starts under 4 years of age. However, some clinical trials did not confirm any efficacy of quinidine in patients with MMFSI and ADNFLE. This precision treatment failure was probably due to inability to reach enough therapeutic concentrations of quinidine in the cerebrospinal fluid. Moreover, the use of quinidine is limited by undesired cardiac effects, particularly QT prolongation [70,71,72,73,74].

The *KCNT2* gene, known also as Slick or Slo2., is found on chromosome 1q31.3 and encodes the KNa1.2 channel subunit. Disorders linked to *KCNT2* mutations are manifested as an overlapping spectrum of DEEs. An example is EIEE, which is typically associated with convulsions (tonic spasms) occurring during first months of life. EIEE is one of the earliest forms of DEE. It is presumed that *KCNA2* variants may be related to EIEE type 57. Another consequence of this mutation may be the development of EIMFS [75,76,77,78].

Known *KCNT2* variants present GoF characteristics. Among the available antiepileptic drugs, phenobarbital, topiramate, valproic acid, oxcarbazepine, and lamotrigine are used in the treatment of DEE. Moreover, quinidine can be considered for treating patients with *KCNA2* mutations. Ambrosino et al. [79] described a patient who responded positively to a combination of quinidine and valproate. Unfortunately, the antiseizure effects of quinidine can disappear over time, as in the described case of the *KCNT2*-related West and Lennox–Gastaut syndromes [3]. Table 2 summarizes the data on epilepsy due to mutations in genes encoding potassium channels.

## 4. Genes Encoding GABA_A_ Receptors

GABA_A_R (GABA receptors) are responsible for inhibitory neurotransmission, comprising five protein subunits surrounding a chloride channel. There are numerous genes encoding for different subunits of the GABA_A_Rs, including six α (α1–6), three β (β1–3), three γ (γ1–3), and the less-common δ, Ε, and π subunits. However, most receptors contain two α, two β, and one γ subunit. The channel within the GABA_A_R complex opens in response to positive stimulation of binding sites, allowing the inflow of chloride ions into neurons. This, in turn, causes membrane hyperpolarization and raises the seizure threshold. Different binding sites within GABA_A_R subunits interact specifically with distinct ligands, including GABA, benzodiazepines, barbiturates, neurosteroids, or convulsants (like picrotoxin). For years, GABA_A_R-targeting drugs have been essential for the treatment of seizures [82,83,84,85].

Since GABA_A_Rs are necessary for inhibitory processes in the brain, the presence of LoF mutations in GABA_A_R subunit genes (*GABRs)* can lead to impaired gaiting of receptors, hyperexcitability, and the development of seizures. Next-generation sequencing has made it possible to relate a variety of conditions to specific GABA_A_R subunit polymorphisms. Presently, correlations between *GABRs* and epilepsy, eating disorders, autism, and bipolar disorders have been documented. As it was reported, 24 non-synonymous *GABRs* variants are seizure-causative in monogenic epilepsies, whereas 3 additional variants were detected in non-monogenic cases. Specifically, monogenic genetic epilepsies may result primarily from mutations in *GABRA1*, *GABRB3*, and *GABRG2*, which encode for the predominant α1β3γ2 GABA_A_ isoforms. Missense or nonsense mutations of *GABRA1*, *GABRB3*, and *GABRG2* are linked with GEFS+, childhood absence epilepsy, febrile seizures, and juvenile myoclonic epilepsy. Mutations in other genes encoding for the α6, β2, or δ subunits of GABA_A_ receptors have been found in either animal models or patients with epilepsy. It is known that Dravet syndrome can be a result of mutations in genes encoding the α1, β1, β2, and γ2 subunits of GABA_A_Rs. A de novo mutation of the β1(F246S) subunit was detected in a patient with infantile spasms, while a de novo mutation in exon 4 of *GABRB2* (β2(M79T)) was identified to cause intellectual disability and epilepsy [82,83,84,85,86].Epilepsy treatment with benzodiazepines or barbiturates, non-specific positive allosteric modulators of GABA_A_ Rs, is quite effective. However, their use is limited by noticeable central undesired effects and a high risk of addiction. Numerous benzodiazepines are employed sporadically or as needed, rather than on a regular schedule. Therapy with other antiepileptic drugs enhancing GABAergic transmission is also considered. An example would be topiramate, which increases the rate of chloride channel opening by binding to functional sites within the β_1_ and β_3_ subunits. Darigabat is a selective allosteric positive modulator that targets the GABA_A_ receptor α 2, 3, and 5 subunits. This drug has demonstrated significant efficacy in several preclinical models of epilepsy (e.g., a mouse model of drug-resistant seizures in mesial temporal lobe epilepsy) and has produced promising outcomes in a clinical trial for photosensitive epilepsy [86].

## 5. Mutations in Genes Encoding Ionotropic Glutamate Receptors

Glutamate is the major excitatory neurotransmitter in the central nervous system. Additionally, it plays the role of a gliotransmitter, modulating synaptic efficacy and regulating the release of various biological compounds, including cytokines. In the cerebral cortex, approximately 70% to 80% of the neurons are glutamatergic, with the remainder comprising GABAergic inhibitory interneurons. Glutamate binds to ionotropic and metabotropic glutamate receptors (iGluRs and mGluRs, respectively). iGluRs are categorized into four groups of receptors: α-amino-3-hydroxy-5-methyl-4-isoxazolepropionic acid (AMPA), kainate, N-methyl-d-aspartate (NMDA), and GluD (delta), which differ in their physiological roles and properties. These receptors are constructed from subunits encoded by 18 genes [87,88].

### 5.1. Mutations in Genes Encoding NMDA Receptors

NMDA receptors (NMDARs) are involved in excitatory synaptic transmission and related processes involving long-term potentiation and memory. NMDARs are heteroterameric ion channels containing two obligatory GluN1 subunits with co-agonist glycine binding sites. Additionally, these receptors contain two GluN2 subunits which incorporated glutamate binding sites, or sometimes one GluN2 and one GluN3 subunit. At resting membrane potentials, NMDARs are blocked by physiologic concentrations of extracellular magnesium ions. Activation of NMDARs requires two molecules of glycine and two molecules of glutamate. Depolarization of the neuron removes the magnesium block and enables the inward movement of sodium and calcium ions. In turn, the entry of calcium ions contributes to a cascade of intracellular events that may trigger long-term potentiation or long-term depression of synaptic currents. Excessive activation of NMDARs can raise intracellular calcium concentrations to supraphysiological levels, running apoptotic processes in neurons. Physiologically, activation of NMDARs produces slower and longer excitation when compared with AMPARs. This long abnormal excitation plays a role in the generation of seizures related to GoF mutations in genes encoding NMDARs, such as *GRIN1*, *GRIN2A*, *GRIN2B*, and *GRIN2D* [87,88,89,90,91,92,93].

Mutations in *GRIN* genes are related to epilepsy and other neurodevelopmental disturbances including autism spectrum disorder, intellectual disability, developmental delay, and schizophrenia. Therefore, NMDARs are becoming an increasingly important targets in the design of new drugs that are potentially effective in the treatment of these disorders. However, despite these efforts, there are still too few effective and safe drugs for neuropsychiatric diseases [89].

Since GluN1 is an obligatory subunit in a functional NMDAR, mutations in the *GRIN1* gene, located on 9q34.3 chromosome and encoding this subunit, seem to have the greatest influence on neuronal activity [91,94]. In fact, mutations in *GRIN1* (comprising 50% of cases of LoF variants) are related to various infantile-onset seizures and EOEE manifested by intractable seizures, severe developmental delay, and intellectual disability [94]. Specifically, one of the LoF *GRIN1* mutations results in neurodevelopmental encephalopathy. On the other hand, GoF *GRIN1* variants are usually related to several seizure phenotypes, including epileptic spasms and tonic, focal, myoclonic, local migrating, or altering seizures. Accompanying EEG changes manifest as suppression bursts, multifocal spikes, hypsarrhythmia, and slow spike waves as well as continuous spikes and waves during slow sleep. Interestingly, mutations in *GRIN1*, *GRIN2B*, and *GRIN2D* present more severe clinical phenotypes than *GRIN2A* variants [88,89,90,91].

Further NMDAR genes associated with epilepsy are *GRIN2A* and *GRIN2B*. Either LoF or GoF mutations in these genes can lead to benign temporal lobe epilepsy, atypical benign partial epilepsy, EE with continuous spikes and waves during slow sleep, and Landau–Kleffner syndrome. In some patients, motor and speech disorders may be observed. Moreover, a GoF *GRIN2B* mutation was reported to cause West syndrome and epileptic encephalopathy [88,89,90,91,92].

Mutations in the *GRIN2D* gene, which encodes NMDA2D receptors, are related to DEE, which is characterized by mild-to-profound developmental delay or intellectual disability, epilepsy, abnormal muscle tone (hypotonia and spasticity), movement disorders (dystonia, dyskinesia, chorea), autism spectrum disorders, and cortical visual impairment. Epilepsy manifestations include focal, generalized, febrile, or absence seizures and epileptic spasms. Patients with these conditions respond well to known NMDA antagonists, e.g., memantine, dextromethorphan, dextrorphan, amantadine, and ketamine [88,91,93].

### 5.2. Mutations in Genes Encoding AMPA Receptors

AMPARs are the main glutamate ionotropic receptors, which are predominantly expressed in the postsynaptic neuronal membrane, mediating fast excitatory synaptic neurotransmission within the brain’s neural networks. Increased density of AMPARs is frequently found in the brain tissue of patients with focal seizures. AMPARs are homo- or heterotetrameric structures comprised of four types of subunits (from GluA1 to GluA4) and are directly activated by glutamate. No other endogenous ligand has been discovered to date. Receptors containing GluA2 are calcium impermeable. They are mainly expressed in excitatory neurons and play an important role in the development of neural circuitry and synaptic plasticity. In contrast, calcium-permeable AMPARs do not contain the GluA2 subunit and are mainly expressed in inhibitory interneurons [88,94,95].

Variants in AMPARs are not as commonly reported as those in NMDARs, but they can result in cognitive decline, autism spectrum disorder, and less frequently, epilepsy. Similarly, the literature provides limited insights into epilepsy related to *GRIA* genes encoding AMPAR subunits. Patients with de novo mutations in the *GRIA2* gene, located on the 4q32.1 chromosome, present with neurodevelopmental disorders, seizures, and DEE. Most *GRIA2* variants have LoF characteristics. Seizures observed in patients were varied and included focal, tonic–clonic, clonic, and tonic convulsions. Bearing in mind that AMPARs containing GluA2 subunits which are located in inhibitory interneurons, it is not surprising that LoF *GRIN2A* mutations lead to decreased function of these receptors and to seizures. On the other hand, GoF *GRIA2* variants may also be related to seizures (tonic convulsions) that, due to their GoF characteristics, can be successfully treated with perampanel (Fycompa), a negative allosteric modulator of AMPARs. Perampanel has been approved as an add-on therapy of partial and primary generalized tonic–clonic convulsions. Interestingly, decanoic acid, a component of the ketogenic diet, can inhibit AMPARs by acting at a site distinct from that of perampanel. Therefore, the two substances act synergistically to provide better seizure control [95,96].

*GRIA3* is an X-linked gene encoding for the GluA3 subunit of AMPARs. Mutations in *GRIA3* may be uncommonly related to epilepsy. Documented cases have been characterized by brief myoclonic and clonic movements or atypical absence seizures without prominent motor manifestations. The drugs used in the treatment of these disorders include midazolam, ketamine, thiopental, and propofol. All of the mentioned medications are GABA_A_R agonists or NMDAR antagonists. Moreover, one patient with daily multiple atypical absences associated with eyelid myoclonia was reported to be treated with topiramate. This antiepileptic drug has a complex mechanism of action, including potentiation of GABA_A_ transmission and inhibition of AMPARs, and it completely freed the patient from major motor seizures [97].

## 6. Discussion

Understanding the complex interplay between ion channels, ionic receptors, and epileptic activity advances our knowledge about the pathogenesis of epilepsy and gives hope for the development of targeted therapeutic interventions. Currently, most epilepsies are diagnosed based on their clinical/EEG manifestations and are assigned to a specific type. The majority of the available antiepileptic drugs enhances inhibitory and attenuates excitatory neurotransmission. Therefore, they usually belong to classes of GABA_A_ receptor agonists, glutamate receptor antagonists, sodium channel blockers, or potassium channel openers. Some antiseizure medications, like valproate, topiramate, or felbamate, have complex mechanisms of action, combining effects on several targets. However, despite many efforts to develop new drugs, 30–50% of epilepsies remain drug resistant. Furthermore, sensitivity to a given drug varies among patients with the same type of epilepsy.

Genetic epilepsies account for approximately 50% of all cases of this neurological disorder. Congenital and monogenic seizures have received increasing attention for over 20 years. It is clear that mutations in a large number of genes—possibly hundreds—can contribute to the development of seizures. Increasing data indicate that identification of gene variants can have significant implications for the diagnosis and treatment of epilepsy syndromes. But again, it turned out that patients with the same mutation can differently response to the applied treatment. While some patients responded well to one antiepileptic drug, others required specific drug combinations [62,97,98]. One of the reasons for the unsatisfyingly poor genotype–phenotype correlation is that epilepsies present a complex inheritance pattern usually involving more than one gene. Epilepsy genes can carry all possible types of mutations. For instance, truncating mutations in *SCN1A* lead to earlier seizures in Dravet syndrome when compared to missense mutations. Furthermore, even the same type of mutation may be manifested in different individuals through seizures of varying severity. As an example, *KCNQ2* mutations can result in either benign BFNS or severe EEs. Additionally, a large number of epilepsy genes, different gene penetration, mosaicism, and the influence of epigenetic factors should be also taken into consideration. This makes genetic classification of epilepsy very difficult. Therefore, identification of the mutant gene cannot be directly translated into a clinical diagnosis [3]. It should also be remembered that seizures are only a part of epilepsy syndromes, and isolated anticonvulsant effects may not improve a patient’s quality of life. Furthermore, a large number of diagnosed syndromes and various types of clinical seizures do not facilitate the development of precision therapies [98].

The majority of sodium channel gene abnormalities have been found in patients with DEE. This disturbance is characterized by uncontrollable seizures and an increased risk of SUDEP. Mutations in the *SCN1A* gene are linked to Dravet syndrome, manifesting as partial or generalized epilepsy with febrile seizures. Since Dravet syndrome is caused by LoF mutations in inhibitory interneurons, sodium channel blockers, including phenytoin, carbamazepine, oxcarbazepine, or lamotrigine, can intensify the convulsions occurring in this condition. In contrast, valproate, benzodiazepines, stiripentol, cannabidiol, and the ketogenic diet show some effectiveness. Currently, a combination of stiripentol, valproate, and clobazam, or alternatively, valproate with topiramate, are approved for the treatment of Dravet syndrome [9,10,11,12,13,14,15,16,17]. It is worth mentioning that stiripentol was found to be more effective in patients with missense *SCN1A* mutations than in those with truncation mutations [1].

A broad spectrum of seizures is associated with *SCN2A* mutations. Most variants lead to epileptic encephalopathies. De novo mutations typically produce more severe phenotypes. *SCN2A* variants may have either GoF or LoF characteristics. Patients with early-onset seizures present missense GoF mutations and respond well to sodium channel blockers, particularly high-dose phenytoin and carbamazepine. A ketogenic diet and high-dose steroid treatments are also effective. On the contrary, seizures occurring during the course of late-onset forms of epilepsy, which are associated with LoF truncation *SCN2A* mutations, worsened during treatment with sodium channel inhibitors [21,22].

Most *SCN8A* mutations causing seizures present GoF characteristics. Therefore, epilepsy related to these variants is controlled by treatment with sodium channel blockers, e.g., phenytoin, carbamazepine, oxcarbazepine, lacosamide, lamotrigine, rufinamide, or topiramate. Two investigational drugs are worth mentioning: NBI-921352 (formerly XEN901), a selective Na_v_1.6 inhibitor, and GS458967, a potent sodium channel modulator [27,28,29,30,31].

*KCNQ2* and *KCNQ3* gene mutations usually lead to BNFE or to more severe epileptic encephalopathies. Refractory seizures are typically caused by de novo mutations, while BNFE-related variants are passed from parents. Since these mutations have predominantly LoF characteristics, patients respond well to retigabine, a selective Kv7 channel activator. For unclear reasons, some sodium channel blockers, including carbamazepine and lamotrigine, may also be effective. Additionally, a ketogenic diet was proved to restore the function of Kv7.3 channels in LoF *KCNQ3* mutations. Among the investigational drugs, XEN496, a novel immediate-release formulation of retigabine, remains under a Phase III clinical trial. Not surprisingly, patients with encephalopathy resulting from GoF *KCNQ3* variations do not benefit from treatment with sodium channel blockers, while retigabine can even aggravate encephalopathy symptoms [40,41,42,43,44,45].

*KCNA1* mutations, which mainly cause EAs, have LoF characteristics. Therefore, they can be treated with sodium channel blockers, such as carbamazepine, phenytoin, and lamotrigine. Positive effects were also reported for acetalozamide and benzodiazepines. An interesting alternative drug is niflumic acid, which may ameliorate channel dysfunctions by enhancing the activity of Kv1.1 channels. *KCNA2* mutations may be either LoF or GoF. LoF variants are related to EEwith a good response to sodium channel blockers, while GoF mutations are less sensitive to this therapy [54,55,56,57,58,59,60,61,62,63,64,65,66].

On the other hand, all known variants of the *KCNT1* gene turned out to have GoF phenotypes. Mutations in this gene cause ADNFLE, MMFSI, EIMFS, and other severe phenotypes in neonates and infants. Seizures related to these syndromes are usually highly drug resistant. A combination of stripenol and benzodiazepines (commonly clonazepam, clobazam, or nitrazepam) or levetiracetam and a ketogenic diet showed very limited therapeutic effects. Theoretically, seizures caused by *KCNT1* mutations can be treated with quinidine, an antagonist of potassium channels. Unfortunately, the response to quinidine is unpredictable and limited by induced cardiotoxic effects. In turn, *KCNT2* variants, causing epileptic encephalopathies, present with GoF characteristics. Phenobarbital, topiramate, valproic acid, oxcarbazepine, and lamotrigine are commonly used in the treatment of these disorders. Quinidine and its combination with valproate may be also effective [54,55,56,57,58,59,60,61,62,63,64,65,66].

Several monogenic epilepsies may result from mutations in the *GABRA1*, *GABRB3*, and *GABRG2* genes encoding for the predominant α1β3γ2 GABA_A_ isoforms. Missense or nonsense mutations of *GABRA1*, *GABRB3*, and *GABRG2* are linked with GEFS+, childhood absence epilepsy, febrile seizures, and juvenile myoclonic epilepsy. However, Dravet syndrome can result not only from *SCN1A* mutations, but also in genes encoding the α1, β1, β2, and γ2 GABA_A_R subunits. Treatment with benzodiazepines or barbiturates is effective, but associated with serious central side effects. Therefore, they are often replaced by topiramate. According to experimental data, an alternative may be darigabat, a selective allosteric positive GABA_A_ modulator [86].

Mutations in *GRIN* genes encoding NMDAR subunits are related to epilepsy, schizophrenia, and neurodevelopmental disorders. In fact, mutations in *GRIN1* are related to a variety of infantile-onset seizures (GoF mutations) and encephalopathies (LoF variants). In turn, either LoF or GoF mutations in *GRIN2A* and *GRIN2B* genes cause benign temporal epilepsy and Landau–Kleffner syndrome. Moreover, one GoF *GRIN2B* mutation was reported to cause West syndrome and epileptic encephalopathy. On the other hand, variations in *GRIN2D*, encoding NMDA2DRs, mainly cause DEE with a variety of convulsions. Importantly, patients with GoF *GRIN* mutations respond to NMDA antagonists (e.g., memantine, dextromethorphan, dextrorphan, amantadine, and ketamine) [92,93].

Finally, mutations in *GRIA* genes, encoding AMPAR subunits, rarely lead to seizures. Nevertheless, patients with de novo LoF *GRIA2* mutations developed DEE associated with different types of convulsions. However, GoF *GRIA2* variants may also result in seizures that, due to their GoF feature, can be effectively treated with perampanel, a negative allosteric modulator of AMPARs. Also the ketogenic diet weakens the function of AMPARs, supporting the action of perampanel. Similarly, mutations in the *GRIA3* gene may be related to epileptic seizures with different morphologies. Drugs used in the treatment of patients with *GRIA* variations usually enhance GABAergic and/or reduce glutamatergic neurotransmission. Common examples include midazolam, ketamine, thiopental, propofol, and topiramate [94,95,96].

It should be mentioned that seizures in genetic epilepsies are drug-resistant in the majority of cases. The effectiveness of a new antiepileptic drug, used as an adjunctive therapy, is considered as a significantly decreased seizure frequency upon application of this drug compared to a control. This rarely translates into an improvement in the patient’s quality of life, particularly in developmental epileptic encephalopathies. Failures of precision therapies should be not underestimated, e.g., quinidine (and its derivatives) in *KCNT1*-activating mutations or phenytoin in patients with GoF *SCN8A* mutations and encephalopathy. Accurate selection of antiepileptic drugs at the beginning of treatment can prevent the use of ineffective drugs and improve the prognosis of patients. For some time now, machine learning algorithms have been available that process multiple datasets, including the duration of epilepsy, epilepsy syndrome, age of onset, cognitive impairment, and gene mutations. These algorithms may be helpful in the design of treatments; however, as in the case of clinical trials, larger sample sizes in such studies are required to increase their reliability [99,100].

The presented data confirm that gene mutations are an increasingly recognized factor in the pathogenesis of epilepsy. Genetic testing may contribute to the diagnosis and treatment of genetic epilepsies. Nevertheless, there is no simple translation between the identification of mutations and the diagnosis or treatment of epilepsy. To fully understand the mechanisms causing epilepsy, we should focus on sequencing huge populations of people. This is especially true since the genotype–phenotype translation depends not only on the type and position of the mutation. Many other issues, like the exact functional changes induced by gene variants or the influence of modifier genes and environmental factors, can contribute to the development of a given phenotype. Therefore, predicting how genes translate into phenotypes is a real challenge. Detailed knowledge about the functions of defective genes (LoF, GoF variants) may facilitate the design of therapies based on available or repurposed drugs. To establish a better relationship between gene mutations and treatment response, a greater emphasis should be placed on the development of preclinical research based on animal models that replicates human syndromes and reflect their complex nature. Unfortunately, due to a multitude of factors that determine the development of epilepsy syndromes, the available models using genetically modified animals do not replicate clinical phenotypes. This, in turn, makes it difficult to find disease-modifying drugs. In many cases, drugs that are effective in experimental conditions are found to be ineffective in clinical practice. Recently, in silico methods have accelerated drug discovery and may be helpful in the search for medications that are effective in genetic epilepsies. Since there is no approved causative treatment for such cases so far, more attention is being paid to gene therapy, particularly in the case of monogenic epilepsies, e.g., Dravet syndrome related to *SCN1A* mutations. The available methods of gene therapy allude to the functional consequences of respective mutations. Gene replacement therapy is intended to restore cellular function in disorders due to LoF variants. Genetic substrate reduction therapy reduces the overproduction of substrates in the case of GoF mutations. Finally, transcriptional enhancement increases the expression of a given gene through the use of different regulatory factors [3,101,102].

## Figures and Tables

**Table 1 ijms-24-16280-t001:** Summary of the phenotypic and genotypic information of epilepsy related to sodium channels.

Gene	Syndrome	Functional Analysis	Type of Seizures	Treatment Used	Recommended Treatment	Reference
*SCN1A*	GEFS+	GoF	febrile	STRP, VPA, CLB, TPM, CBD, STCL, FEN	STRP + VPA + CLB	[9,10,11,12,13,14,15]
	DEE	LoF	Tonic–clonic, myoclonic			
*SCN2A*	BFNIS	GoF	partial, secondary generalized	PHT, CBZ, LEV, BDA, VPA	high-dose PHT	[20,21,22,31]
	DEE	GoF	nd			
	Episodic ataxia	GoF	nd			
	Autism spectrum disorder	LoF	nd			
	Intellectual disability, DEE	LoF	nd			
*SCN8A*	BFNIE, DEE	GoF	tonic	PHT, CBZ, TPM, LEV, VAL, PB	high-dose PHT	[27,28,29,30,31]
	EIEE13	GoF	myoclonic			
	Autism spectrum disorder	LoF	myoclonic			
	Intellectual disability	LoF	tonic, clonic, autonomic			

Treatments: CBD, cannabidiol; CBZ, carbamazepine; CLB, clobazam; FEN, fenfluramine; LEV, levetiracetam; PB, phenobarbital; PHT, phenytoin; STCL, soticlestat; STRP, stiripentol; TPM, topiramate; VPA, valproate. Others: BFNIE, benign familial neonatal epilepsy; BFNIS, benign familial neonatal infantile seizures; DEE, epileptic encephalopathy; EIEE13, early infantile epileptic encephalopathy; GEFS+, genetic epilepsy with febrile seizures plus; GoF, gain-of-function mutation; LoF, loss-of-function mutation; nd, not detected.

**Table 2 ijms-24-16280-t002:** Summary of the phenotypic and genotypic information of epilepsy related to potassium channels.

Gene	Variant	Functional Analysis	Type of Seizures	Treatment Used	Recommended Treatment	References
*KCNA1*	c.781G>A	nd	Focal	VPA	CBZ	[47]
	c.888G>T	GoF	Tonic, focal	4-AP, LEV, OXC, VPA, LCM, PHT, CLB, VGB, KD		[64]
	p.Val368Leu	LoF	Tonic–clonic	OXC, PB	OXC	[65]
	c.1213 C>T	nd	Clonic	PB, CLB, CBZ, ZNS, VPA		[66]
	c.1214C> T	nd	Generalized convulsive	PB; CBZ, LEV, PHT, LTG, VPA	AZA	
*KCNA2*	c.1214C>T	LoF	Focal, myoclonic seizures, FDS, focal motor seizures, secondary generalized tonic–clonic	TPM, OXC, VPA, LEV	AZA	[20]
	c.788T>C	LoF	Myoclonic, myoclonic–atonic	PB, VPA, OXC, LEV, CBZ, CLB, LTG	CLB, LTG	
	c.1214C>T	LoF	Focal, focal dyscognitive, focal motor seizures,	VPA, LEV	VPA, CLB, TPM	
	c.1214C>T	LoF	febrile, focal motor seizures, secondary generalized tonic–clonic	OXC, VPA, CLB, STM, LEV, PRED	LEV	
	c.894G>T	GoF	Generalized tonic–clonic, myoclonic	PB, PHT, VPA, CBZ, LTG, CLB, TPM, OXC, LCM, LEV	Nd	
	c.890G>A	GoF	Generalized tonic–clonic, absences	PMD, VPA, LTG	LTG	
*KCNQ2*	c.1678C>T	nd	Myoclonic seizures	PB, VGB	TPM	[22]
	c.917C>T	nd	Tonic	PB, CLB, LEV	VGB, ZNS, KD	
	c.997C>T	nd	Tonic	PB	TPM	
	c.601C>T	nd	Myoclonic	PB, CLB, LEV, PHT, TPM, VGB	PHT, OXC	
	c.338C>T	nd	Tonic, clonic, autonomic	PB	OXC, high-dose steroids, KD	
	c.773A>T	nd	Tonic, clonic, autonomic	CBD	LEV, KD	
	c.638G>A	nd	Tonic, autonomic	LEV, PB	PGB, LEV, VPA, KD	
	c.629G>A	nd	Tonic, clonic	PB, PHT, VPA, LEV	PHT, OXC	
	c.637C>T	nd	Tonic	VPA, VGB, TPM	LTG	
	c.794C>T	nd	Tonic	CBZ, CLB, LCM	LTG	
	c.1118 + 1G>T	nd	Tonic, clonic	VPA, OXC	VPA, OXC	
*KCNQ1*	c.817C>T	nd	Subsequent tonic–clonic		CBZ	[80]
*KCNT1*	c.1283G>A	nd	FIAM, generalized tonic, generalized tonic–clonic	KD, QUIN, CBD	CLB, LRZ, PB	[69]
	c.2849G>A	nd	Generalized tonic, generalized tonic–clonic, FIANM	KD, CBD	LEV, PB	
*KCNT2*	c.991T>A	nd	Epileptic spasm	PB, TPM	nd	[81]
	c.592C>G	nd	Focal, migrating	VPA, TPM, LTG, NZP	nd	
	c.1690A>T	LoF	focal	VPA, LTG, LEV	nd	[77]
	c.720T>A	nd	Focal seizures, myoclonus, epileptic spasm, tonic, atypical absence	TPM, NZP, LEV, LTG, VBG, ETX, LCM, KD	nd	[75]
	c.569G>A, p.	GoF	Epileptic spasm, nocturnal tonic, and bilateral tonic–clonic	VPA, VBG, RUF, PRED, KD, STM	nd	

Treatments: AZA, acetazolamide; CBD, cannabidiol; CBZ, carbamazepine, CLB, clobazam; ETX, ethosuximide; KD, ketogenic diet; LCM, lacosamide; LEV, levetiracetam; LTG, lamotrigine; NZP, nitrazepam; OXC, oxcarbazepine; PB, phenobarbital; PHT, phenytoin; PMD, primidone; PRED, prednisolone; QUIN, quinidine; RUF, rufinamide; STM, sulthiame; TPM, topiramate, VGB, Vigabatrine; VPA, valproate; 4-AP, 4-aminopyridine; ZNS, zonisamide. Others: FIANM, focal impaired awareness non-motor onset; GoF, gain-of-function mutation; LoF, loss-of-function mutation; nd, not detected.

## Data Availability

The data presented in this study are available on request from the corresponding author.

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
