# Peer review of "Genetic Background of Epilepsy and Antiepileptic Treatments"

_ijms, 2023, doi:10.3390/ijms242216280_

Round 1

Reviewer 1 Report

Comments and Suggestions for Authors

Epilepsy is affecting 40-60 million people worldwide, one third presenting resistant epilepsy.           Epileptic seizures is the result of predominance of excitatory over inhibitory in patients brain.

          This is a very interesting literature review regarding the use personalized antiseizures drugs according to the gene mutations found in different epileptic syndromes.

          Orphan drugs - soticlestat, fenfluramine and ganaxolone – are used in patients with drug-resistant seizures such as - Dravet, Lennox-Gastaut, Dup 15q, PCDH 19-clustering epilepsy; each of this syndromes is related to several gene mutations and have different phenotypes.

          The authors thoroughly analyses different type of mutations of    genes encoding sodium and potassium channels as well as glutamatergic and GABAergic receptors.

          One drug could have opposite effect in patients with different pathogenic mutations of the same gene. Sodium channel blockers could be very effective in patients with  seizures related to  gain-of-function (GoF) mutations in genes encoding sodium channels and in the same time could  aggravate seizures related to loss-of-function (LoF) variants of the same genes

Author Response

Reviewer 1

Epilepsy is affecting 40-60 million people worldwide, one third presenting resistant epilepsy.    Epileptic seizures is the result of predominance of excitatory over inhibitory in patients brain.

          This is a very interesting literature review regarding the use personalized antiseizures drugs according to the gene mutations found in different epileptic syndromes.

          Orphan drugs - soticlestat, fenfluramine and ganaxolone – are used in patients with drug-resistant seizures such as - Dravet, Lennox-Gastaut, Dup 15q, PCDH 19-clustering epilepsy; each of this syndromes is related to several gene mutations and have different phenotypes.

          The authors thoroughly analyses different type of mutations of    genes encoding sodium and potassium channels as well as glutamatergic and GABAergic receptors.

          One drug could have opposite effect in patients with different pathogenic mutations of the same gene. Sodium channel blockers could be very effective in patients with  seizures related to  gain-of-function (GoF) mutations in genes encoding sodium channels and in the same time could  aggravate seizures related to loss-of-function (LoF) variants of the same genes

Response:

Thank you very much for time and attention devoted to the review. I really appreciate it.

Reviewer 2 Report

Comments and Suggestions for Authors

Thank you for asking me to review this potentially interesting manuscript. The authors have provided a narrative review of the advantages of identifying gene mutations in patients with epilepsy. Although the statements made are largely correct, the manuscript would benefit from a more nuanced and critical view of the literature. I highlight areas of the manuscript that are potentially overstated below.

1.       The first line of the abstract is heavily overstated. Despite the identification of approximately 500 gene mutations in children with developmental and epileptic encephalopathies, the outcomes of these children have remained static for many years. The three drugs used as examples have not had a significant overall impact. In addition, it is not obvious that the mutation is the diagnosis given the poor genotype-phenotype correlations described in the abstract i.e. is SCN1A mutation a diagnosis (found in Dravet, GEFS+ febrile seizure….) or is Dravet Syndrome the diagnosis (Multiple gene mutations have been associated with the syndrome). Thus, the bold opening statement should be removed or softened enormously to more fairly represent the current state of genetics in epilepsy. Overall, I would recommend a rewrite of the abstract to be less positive and more balanced.

2.       There is a tension throughout the manuscript between identification of mutations and testing of novel therapies. In most circumstances, knowledge of the mutation in no way informed the choice of medication e.g. in Dravet Syndrome none of fenfluramine, ganaxolone, CBD or soticlestat (or most of the other drugs in section 2.1) are directly targeted to the defined pathophysiology of Dravet which the authors define as LoF of NaV1.1 that impairs the function of inhibitory, but not excitatory, neurons in several brain regions. The authors have conflated the genetic mutation with serendipitous observations that a drug may work without drawing a causation link between the two. At the very least this should be clear in the manuscript. This statement applies to almost all of the examples provided throughout the manuscript.

3.       I would like to see some discussion of why genotype-phenotype correlations are so poor in epilepsy, rather than simply stating it. If the genetic mutation is a major contributor to the phenotype, one might expect much tighter genotype-phenotype correlations. If the variance is not a function of the mutation or variant, then some discussion of what is mediating the non-linear relationship between gene and phenotype would be of interest. I know that this is highlighted as an issue in the conclusion, but no approach to addressing the issue is discussed. This raises a nuanced hypothesis that targeting the mediators rather than the mutation directly could have major benefits.

4.       Although there are some statements about failures of precision therapies, I believe that this deserves a section of its own. Quinidine is no more effective than other AEDs for treatment of ‘KCNT1 disease’ even though the biology suggests that it should be useful. The same is true for everolimus in TSC that has little impact on seizures and no impact on TSC associated neurocognitive disorders. In a typical precision therapy frame this is an unexpected result. The implication is that precision therapies, including gene therapies, may not have a major impact DEEs. Appropriate evaluation of precision therapies is required and there should be equipoise on whether they will work.

5.       Table has a column for ‘effective treatment’. It should be explicit in the manuscript that this is a relative term. Effective treatment in a broad sense would mean seizure freedom (at the least) and recovery of cognitive and behavioral abnormalities. This never happens! Thus, the authors should state that effective treatment as currently formulated in the epilepsy world is a reasonably low bar and not enormously associated with improved quality of life.

6.       The final statement in the manuscript is also overstated. There are many medications that have been tested in genetically modified animals that have failed to lead to clinically used treatments and failures of clinical trials when the drug was effective in the animal model. There are many potential reasons for this including the lack of background genetic diversity in most mice models. Thus, animal models should be one course of action but the authors could also consider the use of computer models etc. A broader conclusion would be beneficial.

I recognize that many of my comments seem nihilistic and I am not suggesting that the authors need to agree with all of these points. However, unless they are invalid (and I do not believe that they are) then it is important that the community does not put all of its eggs in one basket and the limitations of the current approaches should be acknowledged 

Author Response

Responces to the Reviewer 1

Reviewer 2

Thank you for asking me to review this potentially interesting manuscript. The authors have provided a narrative review of the advantages of identifying gene mutations in patients with epilepsy. Although the statements made are largely correct, the manuscript would benefit from a more nuanced and critical view of the literature. I highlight areas of the manuscript that are potentially overstated below.

Thank you, for your constructive criticism. I really appreciate it and did my best to find nuanced view of literature. Well, actually, majority of articles are more than optimistic! Time pressure, however, does not help, therefore, if my correction is not satisfactory, please, indicate the key references.

  1. The first line of the abstract is heavily overstated. Despite the identification of approximately 500 gene mutations in children with developmental and epileptic encephalopathies, the outcomes of these children have remained static for many years. The three drugs used as examples have not had a significant overall impact. In addition, it is not obvious that the mutation is the diagnosis given the poor genotype-phenotype correlations described in the abstract i.e. is SCN1A mutation a diagnosis (found in Dravet, GEFS+ febrile seizure….) or is Dravet Syndrome the diagnosis (Multiple gene mutations have been associated with the syndrome). . In addition, it is not obvious that the mutation is the diagnosis given the poor genotype-phenotype correlations described in the abstract i.e. is SCN1A mutation a diagnosis (found in Dravet, GEFS+ febrile seizure….) or is Dravet Syndrome the diagnosis (Multiple gene mutations have been associated with the syndrome).

The first paragraph of the abstract have been reformulated and its message is much more toned down. Also the second paragraph of the introduction has been mitigated and new sentences have been added (in the beginning of Pg.2). Some considerations about poor genotype-phenotype correlation have been placed in the second paragraph of the discussion. 

  1. There is a tension throughout the manuscript between identification of mutations and testing of novel therapies. In most circumstances, knowledge of the mutation in no way informed the choice of medication e.g. in Dravet Syndrome none of fenfluramine, ganaxolone, CBD or soticlestat (or most of the other drugs in section 2.1) are directly targeted to the defined pathophysiology of Dravet which the authors define as LoF of NaV1.1 that impairs the function of inhibitory, but not excitatory, neurons in several brain regions. The authors have conflated the genetic mutation with serendipitous observations that a drug may work without drawing a causation link between the two. At the very least this should be clear in the manuscript. This statement applies to almost all of the examples provided throughout the manuscript.

In the end of 2.1. subsection, the additional excerpt has been added (Pg. 3)

  1. I would like to see some discussion of why genotype-phenotype correlations are so poor in epilepsy, rather than simply stating it. If the genetic mutation is a major contributor to the phenotype, one might expect much tighter genotype-phenotype correlations. If the variance is not a function of the mutation or variant, then some discussion of what is mediating the non-linear relationship between gene and phenotype would be of interest. I know that this is highlighted as an issue in the conclusion, but no approach to addressing the issue is discussed. This raises a nuanced hypothesis that targeting the mediators rather than the mutation directly could have major benefits.

Considerations about genotype-phenotype correlations have been added to the discussion (the 2nd, 12th and the last paragraphs).

  1. Although there are some statements about failures of precision therapies, I believe that this deserves a section of its own. Quinidine is no more effective than other AEDs for treatment of ‘KCNT1 disease’ even though the biology suggests that it should be useful. The same is true for everolimus in TSC that has little impact on seizures and no impact on TSC associated neurocognitive disorders. In a typical precision therapy frame this is an unexpected result. The implication is that precision therapies, including gene therapies, may not have a major impact DEEs. Appropriate evaluation of precision therapies is required and there should be equipoise on whether they will work.

An explaining sentence has been included in the 3rd paragraph of the 3.3. subunit. This problem has also been mentioned in the discussion.

  1. Table has a column for ‘effective treatment’. It should be explicit in the manuscript that this is a relative term. Effective treatment in a broad sense would mean seizure freedom (at the least) and recovery of cognitive and behavioral abnormalities. This never happens! Thus, the authors should state that effective treatment as currently formulated in the epilepsy world is a reasonably low bar and not enormously associated with improved quality of life.

The term “effective treatment” has been replaced with “recommended”. The more appropriate explanation of “effectiveness” in epilepsy has been explained in the discussion (12th paragraph).

  1. The final statement in the manuscript is also overstated. There are many medications that have been tested in genetically modified animals that have failed to lead to clinically used treatments and failures of clinical trials when the drug was effective in the animal model. There are many potential reasons for this including the lack of background genetic diversity in most mice models. Thus, animal models should be one course of action but the authors could also consider the use of computer models etc. A broader conclusion would be beneficial.

The final statement has been much more mitigated, and the whole last paragraph of discussion has been reformulated.

I recognize that many of my comments seem nihilistic and I am not suggesting that the authors need to agree with all of these points. However, unless they are invalid (and I do not believe that they are) then it is important that the community does not put all of its eggs in one basket and the limitations of the current approaches should be acknowledged 

Well, for me these comments are not nihilistic. I see them as valuable and resulting from the much greater experience of the Reviewer. I am a theoretician in this field, so when I read the literature, it is easier for me to accept the enthusiastic statements of other authors.

I have included additional references to the manuscript:

Dhiman V. Molecular Genetics of Epilepsy: A Clinician's Perspective. Ann Indian Acad Neurol. 2017 Apr-Jun;20(2):96-102

Striano P, Minassian BA. From Genetic Testing to Precision Medicine in Epilepsy.

Neurotherapeutics. 2020;17(2):609-615

Shaker B, Ahmad S, Lee J, Jung C, Na D. In silico methods and tools for drug discovery. Comput Biol Med. 2021 Oct;137:104851

Goodspeed K, Bailey RM, Prasad S, Sadhu C, Cardenas JA, Holmay M, Bilder DA, Minassian BA. Gene Therapy: Novel Approaches to Targeting Monogenic Epilepsies. Front Neurol. 2022 Jun 21;13:805007

Rajiv Mohanraj, Martin J Brodie. Measuring the efficacy of antiepileptic drugs. Seizure  2003 Oct;12(7):413-43.

Yang, S., Wang, B. & Han, X. Models for predicting treatment efficacy of antiepileptic drugs and prognosis of treatment withdrawal in epilepsy patients. Acta Epileptologica 3, 1 (2021)

Reviewer 3 Report

Comments and Suggestions for Authors

The main issue discussed in the article is genetic disorders leading to epilepsy.

The authors provide an excellent overview of the problem. They analyzed gene mutations of sodium and potassium channels, gamma-aminobutyric acid receptors, in genes encoding ionotropic glutamate receptors.

The main conclusion of the article is that epilepsy can be caused by various gene disorders. In each individual case it is necessary to apply original treatment.

The article contains 102 references to literary sources.

There is a small note on the design - the reference numbers are duplicated in the bibliography. This needs to be fixed.

Author Response

Reviewer 3

The main issue discussed in the article is genetic disorders leading to epilepsy.

The authors provide an excellent overview of the problem. They analyzed gene mutations of sodium and potassium channels, gamma-aminobutyric acid receptors, in genes encoding ionotropic glutamate receptors.

The main conclusion of the article is that epilepsy can be caused by various gene disorders. In each individual case it is necessary to apply original treatment.

The article contains 102 references to literary sources.

There is a small note on the design - the reference numbers are duplicated in the bibliography. This needs to be fixed.

Response:

Thank you very much for the review and indicating the mistake in References. Everything has been corrected.

Reviewer 4 Report

Comments and Suggestions for Authors

Pag 2  line 48 unclear sentence

           line 73/74  SCN8A repeated twice

Author Response

Reviewer 4

Pag 2  line 48 unclear sentence

           line 73/74  SCN8A repeated twice

Response:

Thank you very much for time devoted to the review. The sentence has been reformulated and the mistake deleted.

Round 2

Reviewer 2 Report

Comments and Suggestions for Authors

The authors have been very responsive to my previous comments. I think the messages are now much more nuanced and balanced. I have no further concerns

Comments on the Quality of English Language

Some minor editing of english language is required